# The Impact of Medial Meniscal Extrusion on Cartilage of the Medial Femorotibial Joint—A Retrospective Analysis Based on Quantitative T2 Mapping at 3.0T

**DOI:** 10.3390/jcm13226628

**Published:** 2024-11-05

**Authors:** Paul Lennart Hoppe, Moritz Priol, Bernhard Springer, Wenzel Waldstein-Wartenberg, Christoph Böhler, Reinhard Windhager, Siegfried Trattnig, Sebastian Apprich

**Affiliations:** 1Department of Orthopedics and Trauma-Surgery, Medical University of Vienna, Währinger Gürtel 18-20, 1090 Vienna, Austria; moritz.priol@tulln.lknoe.at (M.P.); bernhard.springer@meduniwien.ac.at (B.S.); waldstein@theaurora.at (W.W.-W.); christoph.boehler@meduniwien.ac.at (C.B.); reinhard.windhager@meduniwien.ac.at (R.W.); sebastian.apprich@meduniwien.ac.at (S.A.); 2Department of Biomedical Imaging and Image-Guided Therapy, Medical University of Vienna, Währinger Gürtel 18-20, 1090 Vienna, Austria; siegfried.trattnig@meduniwien.ac.at

**Keywords:** femoral cartilage, osteoarthritis, MRI, T2 mapping, medial meniscal extrusion

## Abstract

**Background/Objectives**: The aim of this study was the investigation of any correlation between medial meniscal extrusion (MME) and T2 relaxation times. Furthermore, the impact of different meniscal morphologies on the femoral cartilage was assessed. **Methods**: Fifty-nine knees of fifty-five patients (twenty-four female, thirty-one male) with a mean age of 33.7 ± 9.2 years and without risk factors for MME or osteoarthritis were examined in a 3.0T MRI. MME was assessed quantitatively in accordance with BLOKS score. T2 maps were calculated from sagittal 2D MESE sequences. The region of interest was defined as the load-bearing cartilage at the medial femoral condyle and analysis was performed on two consecutive slices. T2 values were correlated to MME; furthermore, mean T2 values were compared in different grades of MME. **Results**: T2 values showed a strong correlation with increasing MME (r = 0.635; *p* < 0.001) in an exponential pattern. Analogously, knees with MME ≥ 3 mm showed statistically significant higher T2 values (*p* < 0.001) compared to knees with MME ≤ 2 mm and 2.1–2.9 mm; between the latter two, no differences in T2 values were found. **Conclusions**: T2 values showed a strong correlation with increasing MME. Consequently, MME ≥ 3 mm has a detectable impact on the cartilage of the femur.

## 1. Introduction

The menisci of the knee joint play an important role in kinematics, as they stabilize the joint and enlarge the articular surface. During axial loading of the knee, the menisci convert the compressive forces into hoop-stress, thereby reducing the maximal axial forces while distributing the load to a larger surface. Due to this hoop-stress, the menisci become extruded to some extent, which has to be considered a physiological process [1,2]. Currently, pathological meniscal extrusion is defined as an extrusion exceeding 3 mm measured horizontally between the border of the medial tibial margin to the rim of the medial meniscus on MRI [3].

Pathological medial meniscal extrusion (MME) is closely associated with the presence of osteoarthritis (OA), and common radiological findings such as joint space narrowing may be caused by MME [4]. It is commonly acknowledged that there are other contributing factors to the development of OA like cartilage loss, cartilage defects, and meniscus tears. However, the relative impact of each pathology on OA progression is still unclear [5]. Whether MME leads to OA or OA leads to MME has not yet been studied. Further factors associated with meniscal extrusion are meniscus root degenerations and, especially, meniscus tears, and to some extent also knee malalignment and capsular tears [1].

The MRI provides high-resolution multiplanar images for soft tissue evaluation without ionizing radiation [6]. Standard MRI examinations are performed in a supine position without loading of the knee. Hence, the maximal extent of MME during activity may even be pronounced [7]. Furthermore, early degenerative changes in cartilage cannot be evaluated and quantified in standard MR sequences for joint imaging [8]. For this purpose, examinations on measurements of biomechanical and biochemical characteristics of cartilage have been performed. The water content of cartilage has been shown to relate to the amount of glycosaminoglycans and to structural alterations of the collagen alignment. The technique, known as T2 mapping, quantifies water content and collagen fiber orientation in cartilage, showing increased T2 relaxation times in damaged cartilage matrix [9,10,11]. This technique may help in diagnosing early cartilage degeneration and in demonstrating the efficacy of different therapies concerning cartilage preservation and regeneration [12].

Nevertheless, it has to be considered that T2 relaxation times are highly influenced by age [13], body mass index (BMI) [7,14], and static and dynamic loading of the knee immediately before examination [15,16]. Still, it shows great potential as a diagnostic tool for early-stage cartilage degeneration and, therefore, as a prognostic marker for the individual development of OA. [17,18]

The aim of this study was to evaluate the isolated impact of MME on early cartilage degeneration at the medial femoral condyle. The hypothesis of this study was that MME leads to changes in T2 relaxation times even in healthy knees, indicating early alterations in cartilage. Furthermore, established measurements and extrusion cut-off-values for MME diagnosis were examined regarding their impact on cartilage in T2 mapping.

## 2. Materials and Methods

### 2.1. Patient Population

In this retrospective study, patient charts and the clinical history of 882 patients receiving a standard MRI examination of the knee, including T2 mapping, at the Medical University of Vienna between 2011 and 2019 were reviewed. In order to generate a homogenous patient population without any evidence of major risk factors for arthrosis or even signs of beginning OA, all patients exceeding 55 years, with radiological evidence of arthrosis corresponding to Kellgren–Lawrence > 1, body mass index > 28 kg/m^2^, radiological signs of instability in the knee like ruptures of collateral and/or cruciate ligaments, apparent varus deformity, cartilage defects corresponding to the International Cartilage Repair Society (ICRS) > 2 in the load-bearing area, and with previous surgeries in the joint of interest or diagnosed with rheumatic diseases were excluded. Furthermore, patients younger than 18 years of age were excluded in respect of ethical aspects.

Prior to the MRI scan, all patients gave written informed consent to the usage of their anonymized data for scientific reasons. The study protocol was approved by the local ethics committee of the Medical University of Vienna (1493/2019/10.07.2020).

### 2.2. Magnetic Resonance Imaging

The MRI examinations were conducted using a 3.0 T whole body Magnetom TimTrio scanner (Siemens Medical Solutions, Erlangen, Germany) as part of routine screenings. An 8-channel knee array coil (IN vivo, Gainesville, FL, USA) and a gradient strength of 40 mT/m provided results for analysis. The patients were positioned in the supine position with the knee extended and properly fixated with its joint space in the center of the array coil.

The MRI protocol included various sequences of which the following three were used for data acquisition: 1. sagittal two-dimensional (2D) multi-echo spin-echo (MESE) T2 sequence; 2. sagittal 2D high-resolution proton density (PD) weighted turbo spin echo (TSE) sequence; 3. coronal 2D PD weighted, fat-saturated TSE sequence. In Table 1, the MR sequence parameters are specified.

All MR examinations were performed in the morning between 8 a.m. and 12 a.m., and the patients were encouraged to follow daily routines and activities before examination and to arrive by foot. The time between laying down and the first MRI sequence was 5 min. All patients underwent the exact same MRI protocol, which took about 45 min. Thereby, the time between first unloading and examination was standardized for all patients. Acquisition of T2 maps was performed at the end of the MR examination.

### 2.3. Image and Data Analysis

The morphological evaluation of the acquired MR images was performed on a PACS workstation (Agfa, Ridgefield Park, NJ, USA). For analysis of the medial meniscal extrusion, the coronal 2D PD weighted, fat-saturated TSE sequence was used. The MME was defined as the horizontal distance from the medial tibial margin to the medial margin of the medial meniscus (Figure 1), as predefined by Costa et al. [3]. The slice with the most prominent presentation of the intercondyloid eminence was chosen for measurement of MME, as in this slice the most reliable measurements can be taken, and overestimations can be prevented [19]. Any osteophytes were analytically subtracted from the medial tibial margin. Furthermore, the MME was categorized following the Boston–Leeds Osteoarthritis Knee Score (BLOKS) into four grades (0: <2 mm; 1: 2.1–2.9 mm; 2: 3–4.9 mm; 3: >5 mm) [20].

The region of interest in which osteoarthritis develops first in a case of MME is the main loading zone of the medial femoral cartilage. This zone is assumed to lie between the anterior and the posterior horn of the medial meniscus with the knee in extension. Therefore, T2 relaxation times were analyzed in two consecutive slices, facilitating the sagittal 2D MESE T2 sequence, in which both meniscal horns were depicted. Subsequently, the mean T2 value was calculated. T2 relaxation times were assessed in online reconstructed T2 maps using a pixel-wise, mono-exponential, non-negative least square (NNLS) fit analysis (MapIt, Siemens Medical Solutions, Erlangen, Germany). Special attention was paid not to include any artifacts, synovial fluids, or volume effects in the analysis (Figure 2).

All other morphological changes were ascertained by taking all MRI sequences into account. Here, especially the meniscal morphology, but also the criteria for exclusion like ruptures of the collateral and crucial ligaments, as well as cartilage defects, were analyzed.

All MRI evaluations were performed by a senior orthopedic surgeon with over 10 years of clinical experience, together with a senior radiologist with over 25 years of clinical experience.

### 2.4. Statistical Analysis

Statistical analyses were performed using IBM SPSS Statistics Version 27, 64-bit. Normal distributions were assessed by analyzation of histograms. To detect any differences in the distribution of meniscal morphologies to the BLOKS grades, the chi-squared test was used. For examination of the correlation of MME on T2 relaxation times, the Pearson correlation coefficient was calculated. To detect any differences between groups, the Kruskal–Wallis test with the subsequent Dunn’s post hoc test was used. *p*-values ≤ 0.05 were considered statistically significant.

## 3. Results

After applying the above-mentioned exclusion criteria, a total of 55 (24 female and 31 male) patients were included (Figure 3). Among these, 51 received a one-sided MRI examination, and in 4 patients both knees were examined; therefore, 59 knee joints were included in the analysis. The mean age at examination was 33.7 ± 9.2 years and the mean BMI was 23.3 ± 2.5 kg/m^2^.

In the 59 analyzed knee joints, the median medial meniscal extrusion was 2.4 mm (2.0–3.1 mm). Fourteen (23.7%) of the knees corresponded to “BLOKS grade 0”, and twenty-seven (45.8%) knees were classified as “BLOKS grade 1”. In 18 (30.5%) knees, the MME exceeded 3 mm; 17 (28.8%) of those were a “BLOKS grade 2” lesion and a single (1.7%) knee showed a “BLOKS grade 3” lesion.

With increasing MME, the T2 relaxation times of the femoral cartilage at the medial condyle increased, showing a statistically significant correlation. The executed Pearson correlation coefficient showed a strong correlation for the increase in T2 values with increasing MME (r = 0.635; *p* < 0.001). In a subsequently performed curve fitting, the best-matching results for the correlation of MME and T2 values were obtained for a square root function (Figure 4). The much greater increase in T2 values in further progressed MME was also shown when comparing the median T2 values of each BLOKS grade, where there was a statistically significant increase in T2 values in “BLOKS grade 2” compared to “BLOKS grade 0” and “BLOKS grade 1” (*p* < 0.001). In contrast, no statistically significant difference was found between “BLOKS grade 0” and “BLOKS grade 1” (*p* = 0.139), reflecting the minimal increase in T2 values in knees without or just incipient MME.

The morphology of the medial meniscus in the study population classified on the basis of MRI findings was described as “physiological” in 25 (42.4%) knees and as “degenerative altered” in 28 (47.5%) knees. In six (10.2%) knees, a “torn” meniscus was diagnosed (Table 2). Focusing on the morphology of the menisci in each BLOKS group, no statistically significant difference in the rate of “physiological”, “degenerative altered”, and “torn” menisci could be found (*p* = 0.095). However, five of the six torn menisci were observed in the “BLOKS grade 2” group, while the sixth was classified as “BLOKS grade 1” but also showed an extrusion of 2.9 mm. The other two meniscal morphologies were evenly distributed across each BLOKS group. Comparing the T2 values of each group of meniscal morphology, statistically significantly higher T2 values were found in the group of “torn” menisci (*p* = 0.007). Between the groups “physiological” and “degenerative altered” menisci, no statistically significant difference could be found. The median MME was statistically significantly higher in the group of “torn” menisci (*p* = 0.004). Again, a difference between the groups “physiological” and “degenerative altered” menisci could not be found (Figure 5).

## 4. Discussion

In this study, we were able to find a very strong positive correlation between MME and T2 values of cartilage. Moreover, we found statistically significant higher T2 values in knees with meniscal extrusion exceeding 3 mm, which is considered a cut-off for pathologic lesions [3]. As MME and T2 values were correlated in an exponential pattern in this present study, the acknowledged pathological cut-off at 3 mm may be considered a threshold for a consecutive sudden increase in T2 values.

In various studies concerning other compartments of the knee, it has been shown that an increased T2 value correlates with increasing pathomorphological changes of the cartilage [21,22]. Thus, the increased T2 values in the knees with MME ≥ 3 mm might indicate incipient cartilage degeneration, even before morphological changes can be seen on standard morphological MRI.

In a study by Berthiaume et al. [4] focusing on the progression of cartilage volume decrease in the medial compartment, a significant association with MME was observed. However, no quantitative analysis of MME was conducted in this study, which merely differentiated between no extrusion, partial extrusion, and complete meniscal extrusion. Thus, in conformity with our study, higher MME and increased T2 values may reflect the progression of OA in knees.

In a recent analysis, Svensson et al. [23] criticized the 3 mm cut-off for pathological MME, as the high sensitivity is associated with a low specificity for other features of osteoarthritis. They proposed to set the cut-off at 4 mm for a balanced sensitivity–specificity ratio. Considering that the correlation between MME and T2 values shows an exponential pattern, a discussion about specific cut-off values seems not to be as important. This could lead to a more specific, patient-orientated therapeutical approach considering patient-specific risk factors. In the search for therapies delaying the progression of OA, the accurate evaluation of any MME may aid in the process of decision making.

Some studies aimed to classify physiological T2 values for different locations in the knee joint. Joseph et al. [24] issued a reference database on the basis of 481 healthy patients. They reported the highest T2 values of the knee at the medial femur, reaching up to 42.4 ms in females and 40.8 ms in males for the 95th percentile. In our study population, 51.2% of the knees with an MME ≥ 3 mm showed higher T2 values than 42.4 ms. Another study by Soellner et al. [25] identified a global T2 value of 47.6 ms as the threshold for cartilage degeneration in the knee; they validated this by arthroscopic examinations. In our study, 70.7% of the assumed healthy knees, with an MME < 3 mm, were below this threshold.

Furthermore, we found statistically significantly higher MME and T2 values in knees with “torn” menisci, compared to “physiological” and “degenerative altered” menisci. The increased MME in knees due to torn menisci and the consecutive negative effect on cartilage is well studied [4,26]. Costa et al. [3] found that major meniscal extrusion with ≥ 3 mm was related to meniscal tears in 89%, and specifically related to meniscal root tears in 42%. In this present study we were able to demonstrate an association of “torn” menisci with increased T2 values, indicating an impact on the cartilage matrix even before major morphological damage to the cartilage can be seen. This emphasizes the necessity of early meniscal repair. Current studies on repairs of posterior medial meniscal root tears show promising results concerning functional outcome, mostly analyzed using the Lysholm score [27,28,29]. However, these studies also show a progression of Kellgren–Lawrence grading in 16% of the patients and an ongoing cartilage degeneration in 18% of the patients [30]. Potentially, the T2 relaxation time in the femoral cartilage can be used as a prognostic marker for patients undergoing meniscal root repair, leading to better patient selection.

Despite the good functional outcome after meniscal root repair, often no changes in MME are observed in postoperative MRIs [31,32]. This may be due to the fact that MR examinations are most commonly performed with the patient in a supine position. Hence, the unloaded knee is examined, whereby the meniscal hoop-stress and thereby the MME is reduced [7]. The effect of surgical techniques on the cartilage may be analyzed more precisely based on pre- and postoperative T2 values.

To our knowledge, we were the first to correlate the impact of MME with T2 relaxation times of the medial femoral condyle cartilage. Limitations of this study result from a relatively low number of studied knees. However, due to the strict exclusion criteria, we were able to demonstrate the impact of MME on T2 values in knees without essential risk factors, giving an insight into early pathological alterations. Thereby, we included only one knee with an MME corresponding to “BLOKS grade 3”, which did not allow any precise statistical analysis of this subtype.

All MRI examinations were performed with the knee unloaded, which may have caused an underestimation of MME in our study. However, unloaded MRI examinations are widely used in clinical routine, which allows a translation of the results into clinical practice. Furthermore, we analyzed the medial meniscal extrusion in 2D imaging only; in a three-dimensional system like the knee joint, the resulting measurements may lead to reduced validity. Lately, more precise segmentation techniques quantifying meniscal extrusion in 3D have become available [33,34]. However, with our clinically practicable technique, we aimed for easy implementation into clinical practice. As the MR examinations were performed only once for each patient, predictions on the prognostic value could not be given by this retrospective study. Further studies have to be performed, e.g., to investigate the prognostic role of T2 mapping in surgical approaches for MME.

## 5. Conclusions

We found a strong correlation between medial meniscal extrusion and T2 relaxation times, leading to significantly higher T2 values in meniscal extrusions of ≥3 mm. The aim of this study was met. Furthermore, a significant increase in T2 values and MME in torn menisci indicates the importance of meniscal integrity and the necessity of well-analyzed meniscal repairs.

## Figures and Tables

**Figure 1 jcm-13-06628-f001:**
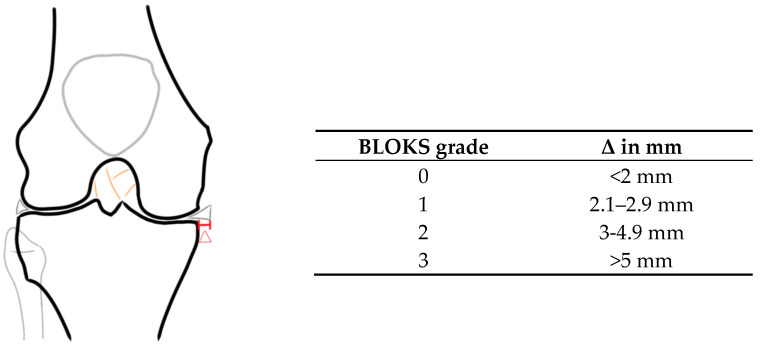
Depiction of measurement of MME, marked in red and labeled with delta, alongside a table of MME categorization according to the Boston–Leeds Osteoarthritis Knee Score (BLOKS).

**Figure 2 jcm-13-06628-f002:**
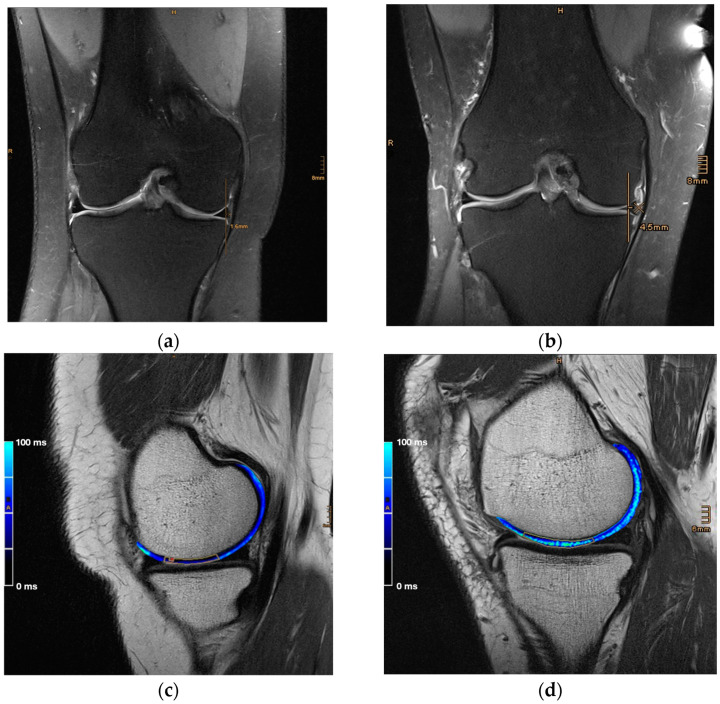
MME measured in coronal 2D PD weighted, fat-saturated TSE sequences and T2 mapping in sagittal 2D MESE T2 sequences illustrated in a healthy knee with a meniscal extrusion of 1.6 mm (**a**) and a mean T2 value of 38.4 ms (**b**), as well as in a pathological knee with a meniscal extrusion of 4.5 mm (**c**) with a corresponding T2 value of 63.9 ms (**d**).

**Figure 3 jcm-13-06628-f003:**
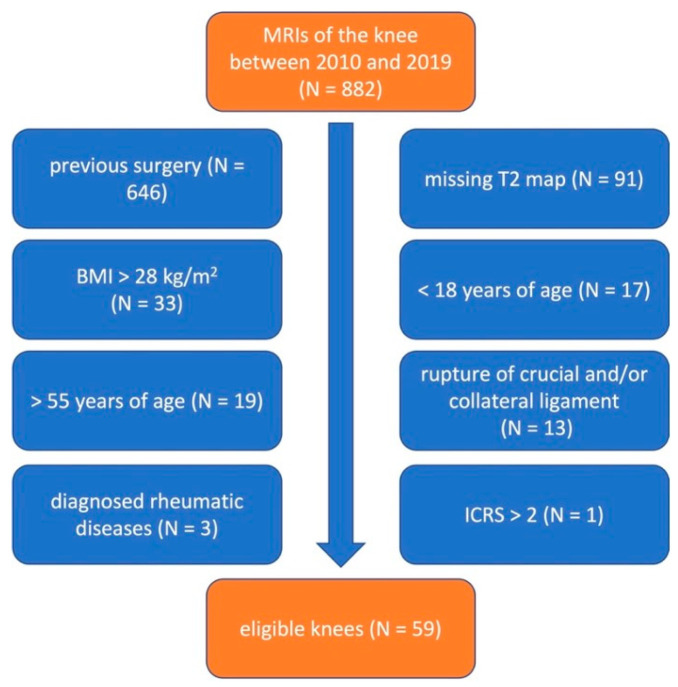
Flow chart of patient selection–exclusion criteria in blue boxes.

**Figure 4 jcm-13-06628-f004:**
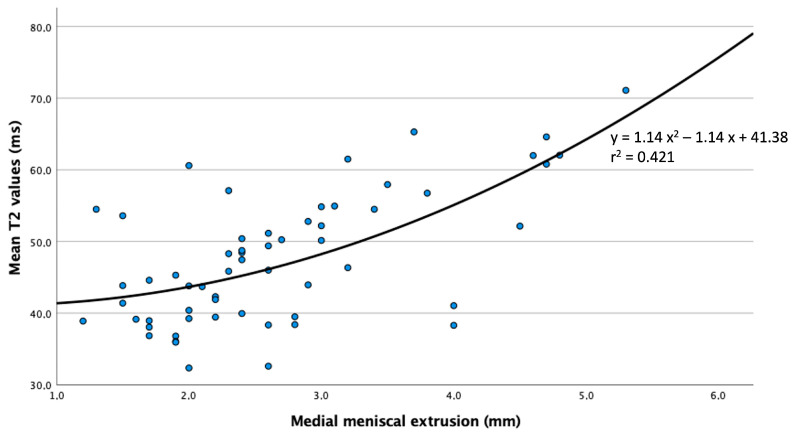
Scatterplot for the correlation of MME and T2 values of the femoral cartilage at the medial condyle showing best concurrence with a square function in curve fitting.

**Figure 5 jcm-13-06628-f005:**
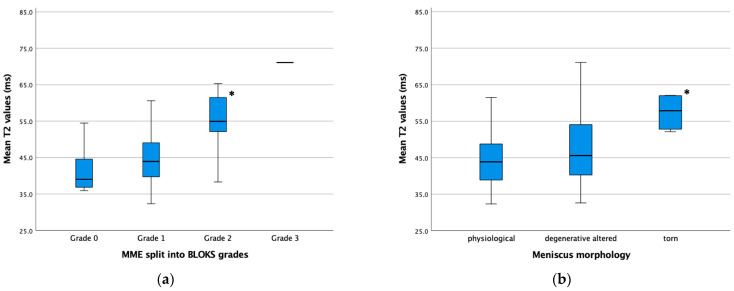
Boxplot illustrating the T2 values of different BLOKS grades (**a**), and T2 values of different meniscal morphologies (**b**); statistically higher T2 values are marked with *.

**Table 1 jcm-13-06628-t001:** Specifications of MRI sequences.

Sequences	Coronal 2D PD TSE	Sagittal 2D TSE	Sagittal 2D MESE
Repetition time (ms)	4250	2130	1200
Echo time (ms)	27	36	6 Echos 11.9–71.4
Field of view (mm)	150 × 150	120 × 120	140 × 140
Matrix	384 × 384	448 × 448	640 × 640
Pixel size (mm)	0.36 × 0.36 × 3	0.22 × 0.22 × 2	0.3 × 0.3

**Table 2 jcm-13-06628-t002:** Meniscal morphologies in each BLOKS group, specified as numbers and percentage of total numbers.

		BLOKS	Grade		
	0	1	2	3	Overall
physiological	6/10.2%	14/23.7%	5/8.5%	-	25/42.4%
degenerative altered	8/13.6%	12/20.3%	7/11.9%	1/1.7%	28/47.5%
torn	-	1/1.7%	5/8.5%	-	6/10.2%
Overall	14/23.7%	27/45.8%	17/28.8%	1/1.7%	59/100%

## Data Availability

The original contributions presented in the study are included in the article; further inquiries can be directed to the corresponding author.

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
