# Peer review of "The Impact of Medial Meniscal Extrusion on Cartilage of the Medial Femorotibial Joint—A Retrospective Analysis Based on Quantitative T2 Mapping at 3.0T"

_jcm, 2024, doi:10.3390/jcm13226628_

Round 1
Reviewer 1 Report
Comments and Suggestions for Authors
overall this is a well conducted and reported study:
-please be sure thar English grammar and syntax are correct, seek aid from a proficient scientific writer if needed
-please be sure that references are correctly formatted and up to date and relevant
-intro section contains some well-known and non relevant information that can be deleted. please report your hypothesis in the end of intro section
-line 79. repost study design
-figure 1 should goin result section togheter with demographic informations
-image analysis: please calculate and report realiability for your measurments
-line 156: I d rather use "correlation" instead of "influence"
-report sample size calculation
-please consider to insert a logistic regression model in your paper
Author Response
Thank you very much for taking the time to review this manuscript. Please find the detailed responses below and the corresponding revisions/corrections highlighted in the re-submitted files.
Comments 1. please be sure thar English grammar and syntax are correct, seek aid from a proficient scientific writer if needed
Response: Thank you for the note. The manuscript has now been revised for accuracy by a native speaker.
Comments 2. please be sure that references are correctly formatted and up to date and relevant
Response: Thank you for the remark. We have re-checked all references and confirm that these are correctly formatted, up to date and relevant.
Comments 3. intro section contains some well-known and non relevant information that can be deleted. please report your hypothesis in the end of intro section
Response: Thank you for pointing this out, we went through the intro section again and deleted the following information (p. 1, l. 45-49 “Specific therapeutical approaches reducing MME especially exist for meniscal root tears, facilitating mainly transtibial pull-out repairs. Good results are reported concerning postoperative functional outcome, however, in some cases cartilage degeneration and OA progresses [6,7].” and p. 1., l. 63-66 “By now, several studies have shown the advantages of T2 mapping compared to other quantitative MR sequences, as it is technically less demanding, can be performed at low field strength and intravenous administration of contrast agent is not required as in comparable techniques like gagCEST imaging or dGEMRIC [10].")
Furthermore, we inserted our hypothesis at the end of the intro section (p. 2 l. 67-69 “The hypothesis of this study was that MME leads to changes in T2 relaxation times even in healthy knees, indicating early alterations in cartilage. “)
Comments 4. line 79. repost study design
Response: Thank you for the improvement, we included the study design into the sentence. (p. 2 l. 73-75 “In this retrospective study, patient charts and clinical history of 882 patients receiving a standard MRI examination of the knee including T2-mapping at the Medical University of Vienna between 2011 and 2019 were reviewed. “)
Comments 5. figure 1 should goin result section togheter with demographic informations
Response: Thank you for pointing this out. We relocated figure 1 and the demographic information into the results section. Consequently, we re-numbered the figures. (p. 5, l. 151-155)
Comments 6. image analysis: please calculate and report realiability for your measurments
Response: Thank you for the suggestion. According to MacKay et al. (2018), “intra-observer, inter-observer and test-retest reliability of compositional techniques were excellent”. Therefore, MRI evaluations were conducted by a senior orthopedic surgeon in consensus with a senior radiologist, no independent MRI evaluation was performed. Therefore, no inter-observer reliability can be reported at this moment.
Comments 7. line 156: I d rather use "correlation" instead of "influence"
Response: Thank you for the remark, we changed the word accordingly.
Comments 8. report sample size calculation
Response: Thank you for the comment. We performed a full survey of all MRIs, which included T2 mapping and were conducted at the Medical University of Vienna. Since this is the first study to examine the correlation between MME and T2 relaxation times, a correlation coefficient could not be anticipated. The post-study power analysis revealed a resulting power of 0.99.
Comments 9. please consider to insert a logistic regression model in your paper
Response: Thank you for the suggestion. However, our main data is ratio-scaled, and transforming one variable into a binary variable would result in a significant loss of information and potential bias in the results.
Reviewer 2 Report
Comments and Suggestions for Authors
I would like to congratulate the authors for this very well presented work. MME is a very interesting topic. Meniscal lessions can lead to arthritis and vise versa.The MRI is the right tool to invetigate these cases and the authors have done a great job depivting the importance of T2 sequences.
I do have some comments for the improvement of this paper
1.Minor English language review. Please use passive tense throughout the manuscript. Please avoid repetitions
2.Lines 88-90 Please avoid being so vage such as "if long legs films were available were analysed" -please delete this.Varus deformity is main cause of medial compartment arthritis and could bias your results. Either state a cut off varus angle that was considered as exclusion criteria or don't mention it at all.
3. Please provide ethics number in the main manuscript , not only in the end.
4. Inter/intra observer reliability between the two main researchers should be reported.
5.Since BLOCKS is the main tool used in this study ,a more thorough description (ie picture) should be provided to help the readers understand it better.
6.Discussion: please delete the fisrt paragraph. Don't repeat your methods. Start this section with your key finding.
7.Conclusions: Rephrase- don't start with "to conclude" . Clearly state your findings and wether the aim of this study was met. Don't finish your paper with the quote "more studies are needed etc".
Author Response
Thank you very much for taking the time to review this manuscript. Please find the detailed responses below and the corresponding revisions/corrections highlighted in the re-submitted files.
Comments 1. Minor English language review. Please use passive tense throughout the manuscript. Please avoid repetitions
Response: Thank you for the note. The manuscript has now been revised for accuracy and re-checked for repetitions by a native speaker. Sentences in active tense have been changed to passive tense throughout the manuscript, except in the discussion and conclusion section.
Comments 2. Lines 88-90 Please avoid being so vage such as "if long legs films were available were analysed" -please delete this.Varus deformity is main cause of medial compartment arthritis and could bias your results. Either state a cut off varus angle that was considered as exclusion criteria or don't mention it at all.
Response: Thank you for pointing this out, the quoted sentence was removed (p. 2, l. 88-90 “Physiological alignment was clinically assessed and documented, if long leg radiographs were available, they were analyzed.”)
Comments 3. Please provide ethics number in the main manuscript , not only in the end.
Response: Thank you for your remark, we provided it in the manuscript accordingly (p. 2 .l. 86 “of the Medical University of Vienna (1493/2019 / 10.07.2020).”)
Comments 4. Inter/intra observer reliability between the two main researchers should be reported.
Response: Thank you for the suggestion. According to MacKay et al. (2018), “intra-observer, inter-observer and test-retest reliability of compositional techniques were excellent”. Therefore, MRI evaluations were conducted by a senior orthopedic surgeon in consensus with a senior radiologist, no independent MRI evaluation was performed. Therefore, no inter-observer reliability can be reported at this moment.
Comments 5. Since BLOCKS is the main tool used in this study ,a more thorough description (ie picture) should be provided to help the readers understand it better.
Response: Thank you for emphasizing the need for an applicable figure, which has now been created and embedded in the manuscript (p. 3, Figure 1). Consequently, we have re-numbered the figures.
Comments 6. Discussion: please delete the fisrt paragraph. Don't repeat your methods. Start this section with your key finding.
Response: Thank you for the comment. We deleted the first paragraph in the discussion to start the section with our key findings as requested. (p. 7, l. 203-207 “we analyzed, for the first time, the impact of MME on T2 relaxation times of the femoral cartilage at the medial condyle in terms of early cartilage degeneration. Due to high influence of different risk-factors on both MME and T2 values, we established a strict approach with rigorous exclusion criteria. In this population, without major distorting factors,”)
Comments 7. Conclusions: Rephrase- don't start with "to conclude" . Clearly state your findings and wether the aim of this study was met. Don't finish your paper with the quote "more studies are needed etc".
Response: Thank you for the improvement, we rephrased the first sentence. (p. 8 l. 273-274 “We found a strong correlation between medial meniscal extrusion and T2 relaxation times, leading to significantly higher T2 values in meniscal extrusions of ≥ 3 mm.“), included that the aim of the study was met (p. 8 l. 274-275 “The aim of this study was met.“) and deleted the last sentence (p. 8, l. 287-289 “Further studies could be performed to investigate the aspect of T2 mapping for usage as prognostic marker in clinical routine.”)